# Genome Analysis of Methicillin-Resistant and Methicillin-Susceptible *Staphylococcus aureus* ST398 Strains Isolated from Patients with Invasive Infection

**DOI:** 10.3390/microorganisms11061446

**Published:** 2023-05-30

**Authors:** Abdeljallil Zeggay, Alban Atchon, Benoit Valot, Didier Hocquet, Xavier Bertrand, Kevin Bouiller

**Affiliations:** 1CHU Besançon, Maladies Infectieuses et Tropicales, 25000 Besançon, France; 2UMR-CNRS 6249 Chrono-Environnement, Université de Franche-Comté, 25000 Besançon, France; 3Bioinformatique et Big Data Au Service de La Santé, UFR Santé, Université de Franche-Comté, 25000 Besançon, France; 4CHU Besançon, Hygiène Hospitalière, 25000 Besançon, France

**Keywords:** human ST398, methicillin-susceptible *Staphylococcus aureus*, CC398, livestock-associated, animals

## Abstract

Background: Using genomic data, we determined the origin of MRSA ST398 isolates responsible for invasive infection in patients with no known livestock contact. Methods: We sequenced the genome of seven MSSA and four MRSA ST398 isolates from patients with invasive infections between 2013 and 2017, using the Illumina technique. Prophage-associated virulence genes and resistance genes were identified. To determine the origin of the isolates, their genome sequences were included in phylogenetic analysis also encompassing the ST398 genomes available on NCBI. Results: All isolates carried the φSa3 prophage, but with variations in the immune evasion cluster: type C in MRSA isolates, and type B in MSSA isolates. All MSSA belonged to the *spa* type t1451. MRSA strains had the same SCC*mec* type IVa (2B) cassette and belonged to *spa* types t899, t4132, t1939 and t2922. All MRSA harbored the tetracycline resistance gene, *tet*(M). Phylogenetic analysis revealed that MSSA isolates belonged to a cluster of human-associated isolates, while MRSA isolates belonged to a cluster containing livestock-associated MRSA. Conclusion: We showed that the clinical isolates MRSA and MSSA ST398 have different origins. An acquisition of virulence genes by livestock-associated MRSA isolates allows them to induce an invasive infection in human.

## 1. Introduction

*S. aureus* belonging to clonal complex 398 (CC398), mainly ST398, took a special place within the *S. aureus* species due to its emergence in the 2000s and its rapid spread in both humans and animals [1]. Within this clonal group, a genome-based analysis revealed the differences between livestock-associated (LA) and human-associated (HA) lineages [2]. LA *S. aureus* is resistant to methicillin due to the presence of the SCC*mec* cassette, to tetracycline due to the presence of *tet*(M) gene, and to zinc due to presence of the *czrC* gene [2]. In contrast, HA *S. aureus* ST398 are mostly susceptible to methicillin and resistant to macrolide due to presence of *erm*(T) gene [3,4]. Another specificity of the HA lineage is the presence of an immune evasion cluster (IEC), supported by the prophage of integrase group 3 (φSa3) [3,5,6]. Thus, the mode of dissemination of *S. aureus* ST398 and clinical data allow for discriminating different groups according to their origin and antimicrobial resistance profile [7]. All sub-populations had an ancestral HA MSSA that was likely prophage-free, from which two sub-lineages emerged and evolved in parallel [2,8]. The LA-MRSA lineage had a livestock reservoir and its evolution was characterized by the absence of HA virulence factors, such as the genes of IEC, and the acquisition of several antibiotic resistance genes such as *mecA*, *tet*(M) and *czrC* [2,7]. The HA-MSSA lineage evolved independently of the cattle reservoir and could easily spread in humans. Its dissemination and evolution were characterized by the acquisition of the IEC that was embedded in the φSa3, and the absence of the *mecA* gene. Through the acquisition of mobile genetic elements, such as φSa3 or φMR11-like-prophage, this lineage is now responsible for severe infection in human without contact with animals. Indeed, LA-MRSA have been mostly reported in localized infections such as skin and soft tissue infections in patients in close contact with livestock, while HA-MSSA has been mainly reported in severe infections such as bloodstream infections (BSI) and bone and joint infections (BJI) [1,9,10]. However, MRSA ST398 isolates of which the genomes harbored φSa3 have been described in China in patients with BSI, which supported the transmission events between animals and humans [11].

In our hospital, all ST398 isolates responsible for BSI between 2010 and 2015 were MSSA [1]. In 2015, we identified cases of infection due to MRSA ST398 in patients with no known contact with livestock [12]. This emergence could be due either to (i) the acquisition by isolates of the LA-MRSA lineage of virulence genes and the prophage φSa3 that promoted spread and infection in human, or (ii) the acquisition of *mecA* gene by isolates of HA-MSSA lineage, already harboring prophage φSa3.

To test these hypotheses, we analyzed the genomes of MRSA and MSSA ST398 isolates responsible for human infections. Genomic comparison, description of virulence factors and antimicrobial resistance determinants allow identification of the origin of these clones.

## 2. Materials and Methods

### 2.1. Isolates Included

We included four isolates of ST398 MRSA isolated in two patients with BJI and two patients with BSI, and seven isolates of ST398 MSSA, randomly selected (from three BJI and four BSI) [12,13]. The isolates were collected between 2013 and 2017 in our hospital. The clinical characteristics of patients, demographic data, past medical history (Charlson comorbidity index), place of residence, and occupation were retrospectively collected from medical charts.

### 2.2. Genome Sequencing and Sequence Analysis

The 11 isolates were sequenced using Illumina NextSeq, which generates 150 bp paired-end reads, with a mean coverage of 80×. The de novo assembly of the genomes of all isolates was performed using SPAdes v.3.11.1. Antimicrobial resistance determinants were identified using Resfinder (version 1.1.4) [14] and The Comprehensive Antibiotic Resistance Database (Card) [15]. These analyses were realized with an alignment greater than 100 bp and a similarity threshold value of 95%. The search for virulence genes was carried out with virulence factor database (Virulence factors of Pathogenic Bacteria, www.mgc.ac.cn/VFs, accessed on 23 June 2021).

### 2.3. Molecular Typing Methods

*Spa* typing was achieved through spatyper (https://cge.cbs.dtu.dk/services/spatyper, accessed on 23 June 2021). SCC*mec* types were determined for the four MRSA genomes with SCCmecFinder (https://cge.cbs.dtu.dk/services/SCCmecFinder, version 1.2, accessed on 23 June 2021) with a threshold at 90% and minimum length at 60%. Prophage was initially detected with the PHASTER software (https://phaster.ca, accessed on 23 June 2021). We further confirmed the presence of φSa3 and φMR11 using in silico PCR with previously described primers [16]. We also investigated the presence of the IEC genes (*sak*, *scn*, and *chp*) by in silico PCR according to Van Wamel et al. [17]. Genomes were annotated with the Prokka program [18]. We extracted the φSa3 sequences from *S. aureus* ST398 genomes with a homemade script, using the biopython library (https://github.com/biopython/biopython, accessed on 21 May 2021). The PGAP (Prokaryotic Genome Annotation Pipeline) program was used for a more detailed annotation of the prophage sequences [19]. The structural comparison of the φSa3 phage sequences was performed using EasyFig [20].

### 2.4. Phylogeny of S. aureus ST398

To understand the population structure of the clone, we downloaded all available *S. aureus* genomes from NCBI in June 2020 (*n* = 11,560). The ST398 genomes (*n* = 861) were then identified by in silico MLST (with a homemade pipeline pyMLST, https://github.com/bvalot/pyMLST, accessed on 25 June 2021) and extracted from this collection. Using genes of a reference genome (CP003045.1), pyMLST identified 2621 core genes further used to build a minimum spanning tree. Isolates of which the genomes had less than 15 alleles of difference were clustered. We kept one isolate per group (*n* = 443) that shared the same *mecA* status, host, and geographic origin. The phylogenetic tree was built with genes present in at least 95% of the strains and with less than 5 mutations per 100 base pairs. The genes were aligned using Mafft (https://mafft.cbrc.jp/alignment/software, accessed on 23 June 2021) and only the regions with SNPs were used, i.e., 15,680 positions. From these data, we built a phylogenetic tree using the Bayesian approach with the GTR + G model using MrBayes (http://nbisweden.github.io/MrBayes, accessed on 23 June 2021). The presence of the *mecA* gene, of prophage φSa3, and the host were annotated using itol (https://itol.embl.de, accessed on 23 June 2021).

### 2.5. Ethics Approval and Consent to Participate

According to French legislation in this period, and because no intervention was performed on patients, no written informed consent was given by the patients. Our study protocol followed the ethical guidelines of the Declaration of Helsinki and was approved by the Institutional Review Board of Besancon Hospital.

## 3. Results

### 3.1. Genome Content and Typing of S. aureus ST398

Table 1 details the content of the 11 isolates ST398 of the collection of genes related to antibiotic resistance, prophages, IEC, and toxins. All *S. aureus* ST398 except one (isolate SaBes 04) carried the ß-lactamase encoding *blaZ* gene. All MSSA harbored the *erm*(T) gene encoding macrolide resistance. No other acquired resistance gene was found among MSSA. MRSA harbored the same cassette IVa (2B)-type SCC*mec* element carrying the *mecA* gene. The other resistance genes detected were *tet*(M), *vga*, and *dfrC* in four isolates, but none of them harbored genes conferring resistance to macrolides or zinc (*czrC* gene). MSSA and MRSA isolates had acquired almost the same virulence genes. Indeed, they all had *cna*, *set* (encoding exotoxin proteins), *fnbpA* and *eta* (encoding exfoliative toxin type A) genes. Five isolates (four MSSA and one MRSA) had *clfA*, but none had *clfB*. None had serin protease genes, enterotoxin, or *tsst* and PVL genes.

### 3.2. Spa Typing and Phylogeny of ST398 S. Aureus Isolates

While the seven MSSA ST398 isolates belonged to *spa* t1451, the four MRSA strains belonged to different *spa* types: t899, t2922, t4132, and t1939 (Table 1). The global *S. aureus* ST398 phylogeny clustered isolates into two sub-lineages (Figure 1). The first contained isolates that were both collected in animals and humans and harbored *mec*A. The second contained mainly strains isolated from humans. In this latter group, a subgroup of *mec*A-negative strains isolated only from humans was distinguishable. The phylogeny analysis of our strains showed that MRSA ST398 isolates belonged to a cluster containing both human and animals isolates that mainly lacked *mec*A, but carried φSa3 (Figure 1A). On the other hand, all MSSA isolates gathered in a cluster containing MSSA strains contained φSa3 isolated in humans (Figure 1B).

### 3.3. Prophage Content of S. aureus ST398 Isolates

Regarding the presence of prophages, no isolates carried the MR11 prophage, but all carried the *int3* gene that indicates the presence of the φSa3 prophage integrated in the usual attachment site (attB) with the core sequence (5′-TGTATCCAAACTGG-3′) in the *hlb* (hemolysin B) gene. Prokka complete genome annotation identified the position of these two fragments having variable sizes. The MSSA strains carried *scn* and *chp* genes corresponding to the IEC type C. The MRSA strains had IEC type B with the presence of *scn*, *chp*, and *sak* genes (Figure 2). Of the eleven genomes sequenced, only the assembly data from five of them (two MSSA, three MRSA) allowed for the extraction of the complete sequence of the φSa3 genome. These genomes were compared to φSa3 prophage genome recovered from NCBI. A comparison of φSa3 genome sequences within the two groups, MSSA and MRSA, showed sequence similarity (Appendix A). Comparison of the genomes of MSSA and MRSA isolates revealed that the φSa3 genome carried by each group differed in size (42,700–43,000 bp) and encoded between 64 and 68 genes. Structural differences were observed at the 5′ end, i.e., in the modules of lysogeny, regulation, replication, and DNA packaging. Regulation genes *dinG* and *ssbA* were only present in MSSA isolates, whereas gene encoding PemK/MazF-like toxin, located next to the integrase in the lysogenic module, was only found in MRSA isolates (Figure 2).

### 3.4. Characteristics of Patients and Types of Infection

The characteristics of patients are shown in Table 2. Seven patients had an MSSA infection and all had comorbidities. Four patients were identified as living in an urban setting, while three were living in a rural setting. Five patients were retired. Concerning the type of infection, four patients had BSI and three had osteomyelitis (two with foreign bodies and one ischiatic pressure ulcer). All bone isolates were obtained from deep tissue samples collected during surgery. No patient died and one relapsed.

Four patients had infection due to MRSA ST398. Two patients had no comorbidity. Three patients lived in a rural setting, whereas one lived in an urban setting. One patient was a farmer, without specifying his exact activity, and had no other specific risk factors for MRSA infection. Two patients had BSI and two had osteomyelitis with foreign body. One patient died due to this infection, and one had complications with several surgeries.

## 4. Discussion

MRSA ST398 was mainly reported in pigs, cattle, and other livestock [7,21]. Previous local data revealed the emergence of MRSA ST398 in 2015, but its prevalence remains below 2% among isolates causing BSI [12]. Although a history of exposure to livestock is the main risk factor for MRSA ST398 human infection, the isolates included in our study were retrieved from patients without identified contact with livestock [22,23]. Whole-genome analysis showed that our isolates had livestock origin, but with the acquisition of mobile genetic elements (i.e., prophage φSa3) conferring the ability to spread and cause severe infection in humans without livestock contact.

According to Price et al., the last common ancestor of *S. aureus* ST398 was probably a human MSSA strain, which at a later stage acquired the SCC*mec* cassette and *tet*(M) gene and lost prophage φSa3, leading to the emergence of LA-MRSA CC398 [2]. However, in a quantitative time-scaled phylogeny, Ward et al. supported the existence of distinct HA and LA clades that both emerged around 1970, with some interspecies transmissions [8]. This was confirmed by the occurrence of transmission found between humans and animals in both directions on several occasions [11].

Phylogeny analysis showed a large genetic distance between MRSA and MSSA ST398 carrying φSa3. This was confirmed by the divergence of the IEC sequence that fosters φSa3. Hence, MRSA of animal origin had an IEC of type B, whereas MSSA of human origin had an IEC type C. These results suggest that LA-MRSA ST398 acquired another φSa3 compared to MSSA ST398, to adapt and spread in humans independently from animal.

Moreover, the presence of the *tet*(M) gene in MRSA isolates from the present collection confirmed that they were derived from the LA clade. Indeed, most of the LA-MRSA ST398 strains isolated in Europe were resistant to tetracyclines and other antibiotics such as gentamicin, ciprofloxacin, macrolides, and lincosamides [24]. The high rate of antibiotic resistance might be the consequence of antibiotics used in livestock (especially tetracycline), but also from the environmental contamination of soils and watersheds [25]. On the other hand, HA-MSSA ST398 isolates were solely resistant to macrolides, and due to *erm*(T) gene, harbored in 80–90% of cases [9].

In practice, it could be important to separate the strains belonging to the ancestral LA-MRSA ST398 clade, because of the importance to health public policy to evaluate and limit the transmission of bacteria between animals and humans. Whole-genome analysis is crucial to determine the origin of *S. aureus* ST398 isolates and differentiate HA-MRSA from LA-MRSA ST398. Indeed, different clades have been identified in *S. aureus* ST398 strains, depending on the epidemiology of *S. aureus* in the country. In China, MRSA isolated from BSI had IEC genes (*sak* and *scn*), but were classified in the human clade in the phylogeny analysis. Moreover, two MSSA strains had no IEC genes, but were classified closely to the MRSA and human clade [11]. These findings suggest the evolution and selection of these clones to facilitate infection in humans. However, the spread of LA-MRSA ST398 harboring φSa3 in the community in patients without contact with livestock appears to remain low. The prevalence for MRSA spa-CC011 (mainly CC398) carrying the φSa3 prophage varied from 1.1% to 18%, depending on the strain origin (more frequent in humans than animals, but more frequent in patients with contact than without contact with livestock) [26,27,28]. The proportion of φSa3-carrying isolates slightly increased from 1.1% (2000 to 2006) to 3.9% (2007 to 2015) in MRSA spa-CC011 isolates from patients in a pig-farming-dense area in Germany. However, Sieber et al. reported a decrease in the proportion of IEC-positive isolates in patients with indirect livestock contact, community-onset, and healthcare-associated community-onset infection compared with patients who had direct contact with livestock in Denmark [26]. These findings suggested that φSa3-positive LA-MRSA could not independently disseminate widely in the general population.

In our study, MRSAST398 isolates belonged to *spa* types t2922, t1939, t899, and t4132, were rarely reported in humans [29,30]. The low proportion of these four *spa* types in humans could explain the low prevalence of MRSA ST398 in our hospital. Interestingly, *spa* type t899 originally belongs to CC9, the predominant LA-MRSA genotype reported in Asia. More recently, it was also reported in a CC9/CC398 hybrid strain in Europe [31,32]. In a Danish study investigating CC9/CC398 isolates collected in humans (colonization or infection), 10/12 isolates harbored the φSa3 prophage and 7/12 clustered with isolates from poultry and poultry meat originated from French production facilities [30]. While the foodborne transmission probably plays a minor role in the epidemiology of LA-MRSA, these results raise the question of the role of poultry meat in the transmission of MRSA ST398 [33].

The prophage φSa3 had a specific role in the virulence and the spread of *S. aureus* CC398 in humans. A previous study revealed several natural alternative integration sites (genes or intergenic regions) of the prophage φSa3 in the genome [27]. In our study, prophage φSa3 was inserted in the same insertion site (i.e., *hlb* gene) in both MSSA and MRSA genomes. Although host specificity is a multifactorial property, previous studies using whole-genome sequencing reported that mobile genetic elements present in ST398 strains were specific for each sub-population and allowed for the differentiation of strains that had a human origin compared to those that had animal origin [11,24,34]. The φSa3 prophage confers an advantage to human colonization and infection [3,28]. Indeed, it included IEC that helps the immune response escape, including the genes *sea* or *sep* (encoding enterotoxins A and P), *sak* (encoding staphylokinase), *chp* (encocoding hemotaxis inhibitory protein), and *scn* (encoding complement inhibitor) [2,22,35]. Few studies have been dedicated to elucidating the regulatory systems of *S. aureus* phages. Consequently, the role of most of the gene products impacting the phage life cycle remains elusive.

While the majority of the ST398 clade lacks the genes encoding toxic shock syndrome toxin or leukocidins (PVL) [2,10], *S. aureus* ST398 strains with PVL genes have been reported in China and more recently in Australia [36,37]. PVL-positive MRSA ST1232-V strains belonging to CC398 were endemic in the community in China [38]. In Europe, these strains were retrieved from individuals originating from Southeast Asia or with a recent travel history in Southeast Asia [23,37,39]. Recently, five cases were reported in Australia, with only two epidemiologically linked to Asia, suggesting that this strain spread in the Australian community independently from China [36].

Our study had some limitations. Firstly, we had few clinical isolates, especially MRSA ST398. However, we compared our isolates with others published in NCBI. Secondly, long-read sequencing data were not analyzed; however, the selection method of isolates downloaded from NCBI permits for a large comparison of genomes with reliable data. This study focuses only on virulence, resistance genes, the presence or absence of prophage, and genetic difference in the φSa3 prophage. The φSa3 prophage is an origin-specific marker of *S. aureus* ST398 and the cornerstone of the virulence of the bacterium. Finally, the clinical data of strains causing infections and sequenced, were retrospectively collected; thus, exposure data are limited. Indeed, one patient (patient 7) was a farmer with no data on his specific occupation (e.g., the presence of pig or poultry farming). This is very interesting because no other risk factors for MRSA infection were found. Therefore, we cannot totally exclude the possibility of contact with livestock in this patient.

## 5. Conclusions

In conclusion, using phylogenetic analysis, we found that the MRSA ST398 strains emerging in our hospital were very likely derived from LA-MRSA, but possessed virulence genes that could explain their adaptation to humans. Specific surveillance of this strain seems important in the future, to avoid the spread of MRSA ST398 isolates in our hospital.

## Figures and Tables

**Figure 1 microorganisms-11-01446-f001:**
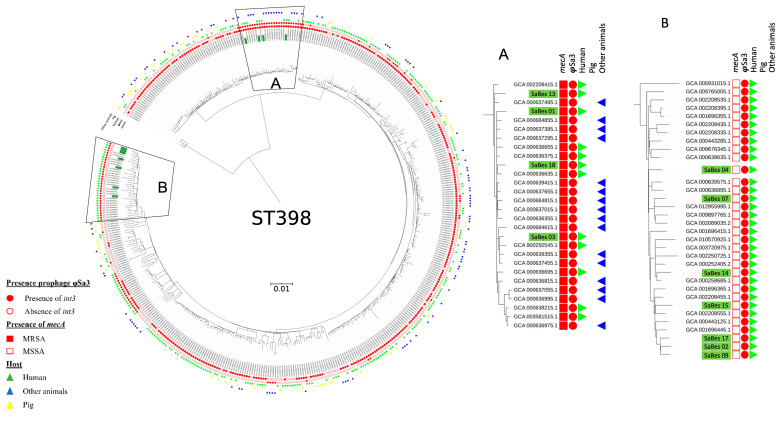
Phylogenetic tree using the Bayesian approach of 443 *S. aureus* ST398 from NCBI and the 11 isolates included in the current study (isolates highlighted in green). MRSA isolates were clustered with animal and human isolates (**A**), whereas MSSA isolates were clustered with only human isolates (**B**). Eleven isolates in the present study harbored φSa3 prophage.

**Figure 2 microorganisms-11-01446-f002:**
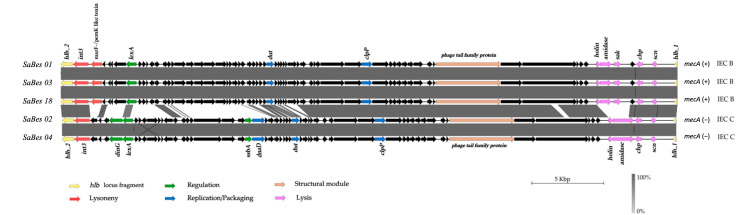
Structural comparison of the genomes of φSa3 prophages in ST398 MSSA and MRSA isolates. The red, green, blue, purple, and orange arrows indicate the different functional modules of the prophage. The immune evasion cluster is included in the lysis module in purple. All MSSA and MRSA isolates, respectively, had the same φSa3 prophages. Only the genome of φSa3 prophage of three MRSA and two MSSA are shown.

**Table 1 microorganisms-11-01446-t001:** Spa typing, presence of SCCmec, virulence and resistance genes.

Isolate	*spa* Type	SCCmec	Resistance Genes				Prophages		IEC Genes		
			*mecA*	*blaZ*	*erm* (T)	*tet* (M)	Prophagic elements (*n*)	*φSa3*	*scn*	*sak*	*chp*
SaBes 01	t4132	IVa (2B)	+	+	−	+	3	+	+	+	+
SaBes 02	t1451	NA	−	+	+	−	2	+	+	−	+
SaBes 03	t899	IVa (2B)	+	+	−	+	2	+	+	+	+
SaBes 04	t1451	NA	−	−	+	−	2	+	+	−	+
SaBes 07	t1451	NA	−	+	+	−	3	+	+	−	+
SaBes 09	t1451	NA	−	+	+	−	5	+	+	−	+
SaBes 13	t1939	IVa (2B)	+	+	−	+	3	+	+	+	+
SaBes 14	t1451	NA	−	+	+	−	3	+	+	−	+
SaBes 15	t1451	NA	−	+	+	−	5	+	+	−	+
SaBes 17	t1451	NA	−	+	+	−	5	+	+	−	+
SaBes 18	t2922	IVa (2B)	+	+	−	+	2	+	+	+	+

NA: not applicable; +: present; −: absent; IEC: immune evasion complex.

**Table 2 microorganisms-11-01446-t002:** Characteristics of patients.

Patient	Isolate	*mecA*	Age (y)	Sex	CCI	Sampling Date	Type of Infection	HAI	Source of Infection	Complications	Setting	Profession	Hosp. in the Last 3 Months
Patient 1	SaBes 01	Yes	21	M	0	2013	OM	Yes	Skin	-	Rural	NA	Yes
Patient 2	SaBes 02	No	60	M	4	2013	Pressure ulcer stage IV	Yes	Skin	-	Rural	retired	Yes
Patient 3	SaBes 03	Yes	77	F	5	2015	OM	Yes	Skin	Multiple surgeries	Urban	retired	Yes
Patient 4	SaBes 04	No	58	M	7	2015	OM	Yes	Skin	-	Rural	NA	Yes
Patient 5	SaBes 07	No	74	F	4	2017	OM	Yes	Skin	Relapse	Urban	retired	No
Patient 6	SaBes 09	No	48	F	3	2017	BSI	No	Skin	Endocarditis, arthritis	Urban	NA	No
Patient 7	SaBes 13	Yes	49	F	0	2016	BSI	NA	Unknown	-	Rural	farmer	No
Patient 8	SaBes 14	No	83	F	5	2016	BSI	No	Unknown	-	Rural	retired	No
Patient 9	SaBes 15	No	83	M	7	2016	BSI	No	Skin	-	Urban	retired	No
Patient 10	SaBes 17	No	71	M	7	2015	BSI	Yes	Skin	-	Urban	retired	Yes
Patient 11	SaBes 18	Yes	86	M	6	2015	BSI	No	Unknown	Death	Rural	retired	Yes

BSI: blood stream infection; CCI: Charlson’s comorbidity index; F: female; HAI: healthcare-associated infection (>48 h after admission); Hosp.: hospitalization; M: male; NA: not available; OM: osteomyelitis.

## Data Availability

Sequencing data are available in the NCBI BioProject PRJNA967250.

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
