# Peer review of "Genome Analysis of Methicillin-Resistant and Methicillin-Susceptible *Staphylococcus aureus* ST398 Strains Isolated from Patients with Invasive Infection"

_microorganisms, 2023, doi:10.3390/microorganisms11061446_

Round 1
Reviewer 1 Report
The authors used whole genome sequencing and phylogenetic analysis approaches to characterize hospital isolates of ST398 Staphylococcus aureus strains emerging in their hospital. The studies suggested that the MRSA ST398 isolates very likely derived from LA-MRSA, but possessed virulence genes that contributed to their adaptation to human hosts. The major limitation of this study, as acknowledged by the authors, is the very small sample size of their ST398 clinical isolates (7 MSSA isolates and 4 MRSA strains). This small sample size coupled with the single site geographical source of strains, allows for limitations on the conclusions that can be stated with any certainty. One of the strengths of this manuscript is the Discussion section, where the authors present their findings in the context of observations from other researchers in the field. Most papers describing virulence factor content of staphylococcal strains fail to present their results in a larger geographical or temporal context. Therefore, even with the small sample size, this work makes a valuable contribution to the field.
Specific Comments:
1. Table 1 seems odd to me. There are four columns that are negative for all of the strains examined (CzrC, MR11, tsst, and lukS). It would be simpler to simply state in the text that all isolates were negative for these. LukS is not mentioned other than in the Table, so it is unclear why it is highlighted here (as opposed to many other virulence or resistance determinants that were not present in any of the strains. The majority of ST398 strains are tst-negative, so why include this in the table when it can simply be stated in the text? Was nothing unique and interesting turned up in the whole genome analysis of these strains?
2. Lines 339-340: This is a circular argument that suggests that it is the spa type that limits human infections. I know of no evidence that indicates that spa type plays a direct role in host range. Host specificity is clearly a multifactorial property.
There were minor grammatical issues. Most just not adding an "s", such as "infection in human."
Author Response
Comment 1. Table 1 seems odd to me. There are four columns that are negative for all of the strains examined (CzrC, MR11, tsst, and lukS). It would be simpler to simply state in the text that all isolates were negative for these. LukS is not mentioned other than in the Table, so it is unclear why it is highlighted here (as opposed to many other virulence or resistance determinants that were not present in any of the strains. The majority of ST398 strains are tst-negative, so why include this in the table when it can simply be stated in the text? Was nothing unique and interesting turned up in the whole genome analysis of these strains?
Answer: We removed the four columns in table 1 and added results in the text. Whole genome analysis did not reveal specific genes in these isolates. We decided to focus genome analysis on φSa3 prophage considering its importance in the spread and virulence of S. aureus ST398.
Comment 2. Lines 339-340: This is a circular argument that suggests that it is the spa type that limits human infections. I know of no evidence that indicates that spa type plays a direct role in host range. Host specificity is clearly a multifactorial property.
Answer : We agree with reviewer and modified the sentence as follow :
« Although host specificity is a multifactorial property, previous studies using whole genome sequencing reported that mobile genetic elements present in ST398 strains were specific for each sub-population and allowed to differentiate strains that had human origin to those had animal origin »
Comment 3. There were minor grammatical issues. Most just not adding an "s", such as "infection in human."
Answer : Corrections have been done.
Author Response
Dear editor and reviewer,
We thank you for the review of our manuscript, entitled “Genome analysis of methicillin-resistant and methicillin-susceptible Staphylococcus aureus ST398 strains isolated from patients with invasive infections“.
We are very grateful for the opportunity to address the reviewers’ concerns and comments. Below are our responses to their critiques and questions. We hope that they will meet your expectations, and we thank the reviewer for helping us to improve the quality of the manuscript.
Comment 1. The article's main limitation is the small number of samples. Despite sequencing 11 strains, only five strains with φSa3 were fully assembled. Relying on just five strains (2 MSSA, 3 MRSA) weakens the argument. While the author compared their findings with genomes downloaded from NCBI, the differences in sources make it unsuitable for drawing conclusive statements. Considering the popularity of long-read sequencing, it is recommended for the author to select representative strains and complete the whole genome analysis.
Answer:
We agree with reviewer that the main limitation is the small number of isolates. Genomes downloaded from NCBI with less 15 different alleles were clustered to avoid redundance of deposit strains and only isolates with mecA status, host, and geographic origin were included to have the most reliable data. Unfortunately, we don’t think that added data on long-read sequencing improve the interpretation of data given the small number of isolates.
We added in the limitation section:
“Long-read sequencing data were not analyzed, however the selection method of isolates downloaded from NCBI permit a large comparison of genomes with reliable data. “
Comment 2. The author's focus on φSa3 alone may be limited. It is suggested that the author utilize pan-genome tools to explore other potential differences between these animal and human ST398 strains. This would provide a more comprehensive understanding of the topic.
Answer :
The reason we focus on φSa3 is the important role of the prophage in the spread and virulence of S. aureusST398. Many studies have explored the S.aureus ST398 genome and the main differences between animal and human isolates were the presence or absence of prophage φSa3. In addition, when prophage is absent in human ST398 strains, patients are most often exposed to close contact with animals (especially pigs).
We added in the limitations section:
"This study focuses only on virulence, resistance genes, presence or absence of prophage, and genetic difference in the φSa3 prophage. The φSa3 prophage is an origin-specific marker of S. aureus ST398 and is the cornerstone of the virulence of the bacterium."